# Detection of *Mycobacterium tuberculosis* complex field infections in cattle using fecal volatile organic compound analysis through gas chromatography-ion mobility spectrometry combined with chemometrics

Pablo Rodríguez-Hernández,[1] María José Cardador,[2] Rocío Ríos-Reina,[3] José María Sánchez-Carvajal,[4] Ángela Galán-Relaño,[5] Francisco Jurado-Martos,[6] Inmaculada Luque,[5] Lourdes Arce,[2] Jaime Gómez-Laguna,[4] Vicente Rodríguez-Estévez[1]

**ABSTRACT** Bovine tuberculosis is considered a re-emerging disease caused by different species from the *Mycobacterium tuberculosis* complex (MTC), important not only for the livestock sector but also for public health due to its zoonotic character. Despite the numerous efforts that have been carried out to improve the performance of the current *antemortem* diagnostic procedures, nowadays, they still pose several drawbacks, such as moderate to low sensitivity, highlighting the necessity to develop alternative and innovative tools to complement control and surveillance frameworks. Volatilome analysis is considered an innovative approach which has been widely employed in animal science, including animal health field and diagnosis, due to the useful and interesting information provided by volatile metabolites. Therefore, this study assesses the potential of gas chromatography coupled to ion mobility spectrometry (GC-IMS) to discriminate cattle naturally infected (field infections) by MTC from non-infected animals. Volatile organic compounds (VOCs) produced from feces were analyzed, employing the subsequent information through chemometrics. After the evaluation of variable importance for the projection of compounds, the final discriminant models achieved a robust performance in cross-validation, as well as high percentages of correct classification (>90%) and optimal data of sensitivity (91.66%) and specificity (99.99%) in external validation. The tentative identification of some VOCs revealed some coincidences with previous studies, although potential new compounds associated with the discrimination of infected and non-infected subjects were also addressed. These results provide strong evidence that a volatilome analysis of feces through GC-IMS coupled to chemometrics could become a valuable methodology to discriminate the infection by MTC in cattle.

**IMPORTANCE** Bovine tuberculosis is endemic in many countries worldwide and poses important concerns for public health because of their zoonotic condition. However, current diagnostic techniques present several hurdles, such as low sensitivity and complexity, among others. In this regard, the development of new approaches to improve the diagnosis and control of this disease is considered crucial. Volatile organic compounds are small molecular mass metabolites which compose volatilome, whose analysis has been widely employed with success in different areas of animal science including animal health. The present study seeks to evaluate the combination of fecal volatilome analysis with chemometrics to detect field infections by bovine tuberculosis (*Mycobacterium tuberculosis* complex) in cattle. The good robust performance of discriminant models as well as the optimal data of sensitivity and specificity achieved highlight volatilome analysis as an innovative approach with huge potential.

Address correspondence to Jaime Gómez-Laguna, j.gomez-laguna@uco.es.

The authors declare no conflict of interest.

See the funding table on p. 15.

**KEYWORDS**   bovine tuberculosis, feces, gas chromatrography-ion mobility spectrometry, mycobacteria, volatile metabolites, chemometrics

M*ycobacterium tuberculosis* complex (MTC) constitutes a wide group of slow-growing mycobacteria species where mainly *Mycobacterium bovis* (and others such as *Mycobacterium caprae*) stands out as the etiological agent for bovine tuberculosis (bTB) (1, 2), as well as for animal tuberculosis due to its multi-host nature (3). bTB is considered an infectious and chronic disease which affects not only different mammal species but also human beings, highlighting its zoonotic character (4, 5). Although human tuberculosis is mostly caused by *Mycobacterium tuberculosis*, about 30% of these cases have been related to *M. bovis* (zoonotic tuberculosis), especially in developing countries with significant livestock bTB prevalence (6–8). Apart from the above-mentioned danger to public health, animal tuberculosis has also a great economic impact in areas such as agriculture, wildlife, and trade (9). This disease reduces the productivity and sustainability of cattle farms due to the compulsory slaughter of infected animals, especially in extensive livestock systems where wildlife reservoirs give rise to complex epidemiological situations (10).

Although major efforts have been made to develop control and surveillance programs in order to eradicate bTB, these are frequently considered inadequate or insufficient. In this regard, some drawbacks associated with current diagnostic techniques and their poor performance, together with other epidemiological and geographical aspects, have been pointed out as responsible for the difficulty in achieving the objective of disease control, especially in endemic countries (10, 11). This context places bTB diagnosis under the spotlight: its time-consuming and tedious features, the lack of enough sensitivity (SE), the specific facilities and high expertise required by the current methodologies, as well as the influence of different variables on the final performance are limitations considered to be of crucial importance (12, 13). Therefore, the optimization of alternative and reliable diagnostic methodologies to avoid such shortcomings has been considered by several authors as the key for bTB control on livestock farms (14).

Volatilome is defined as the group of volatile organic compounds (VOCs) produced by a specific sample or organism, a term framed under volatilomics (15). These substances are small molecular mass metabolites which are generated directly to the environment, offering different advantages such as minimum and easy sample handling and sampling, and non-invasive and direct monitoring (16, 17). Therefore, volatilome analysis has been widely employed for different purposes and applications in animal science (18–21), including animal health and infectious diseases diagnosis (22, 23). In this regard, the potential of volatilome analysis as an innovative diagnostic tool against mycobacteria infection in livestock has recently been highlighted, as well as its several strengths such as the possible development of antemortem, portable, non-invasive methods able to improve the SE and specificity (SP) of the diagnosis (24). VOCs from different biological samples such as serum, feces, or breath have been successfully used to study and diagnose mycobacterial infections in various animal species (22, 23, 25, 26). However, some difficulties due to its initial phase and the scarce number of *in vivo* assays available have been identified. Regarding the infection of *M. bovis* in cattle, there are only a scarce number of studies published in which experimental infections and controlled conditions prevail, and the exhaled breath is the main sample evaluated (24).

Thus, considering the benefits of this approach as well as the advantages that it would provide to the current diagnostic panorama of mycobacteria infection in livestock, the improvement of the state of knowledge and the number of *in vivo* assays of this VOC analysis application is considered important. In recent years, the combination of gas chromatography coupled to ion mobility spectrometry (GC-IMS) has made a breakthrough in the field of metabolomics due to its advantages of high SE, convenient operation, and low cost. In this regard, the goal of the present study was to evaluate and validate the combination of the GC-IMS technique and chemometrics to discriminate naturally infected cattle by MTC from non-infected animals. For this purpose, feces have

been selected as the sample of choice because of the interesting information which can be conveyed from the individual physiology (20) and their benefits compared with other biological matrices for the present goal (15, 24).

## MATERIALS AND METHODS

### Animals and classification criteria

This study is part of a large project on cattle subjected to the surveillance and monitoring for bTB in the framework of the Spanish national eradication program (27). A total of 31 adult dairy cows were included in the present study, belonging to 16 different dairy herds located in northern Andalusia. Two groups of animals were differentiated in terms of MTC infection: 19 cattle were diagnosed as infected (positive), and the remaining 12 individuals were diagnosed as non-infected (negative). Although the majority of non-infected animals were reared in herds with an officially tuberculosis-free status, some of them came from positive farms. The diagnosis of the above-mentioned animals was performed through the bacteriological culture of retropharyngeal, tracheobronchial, and mesenteric lymph node tissue samples and a real-time polymerase chain reaction (real-time PCR) targeting IS*6110* (27). The animals included in the study were considered infected when at least one lymph node tested positive to both procedures.

Briefly, the microbiological culture consisted of a decontamination process of the tissue homogenates with an equal volume of 0.75% (wt/vol: 1/1) hexadecyl pyridinium chloride solution in agitation for 30 min (28). This was followed by a sample centrifugation process of 30 min at $1,500 \times g$, collecting afterwards the pellets with swabs and cultured in liquid media (MGIT 960, Becton Dickinson, Madrid, Spain) using an automatized BD Bacter MGIT System (Becton Dickinson).

A real-time PCR was run after the microbiological culture. In short, specific primers (IS*6110*-forward: 5′-GGTAGCAGACCTCACCTATGTGT-3′; IS*6110*-reverse: 5′-AGGCGTCGG-TGACAAAGG-3′) and probe (IS*6110*-probe: 5′-FAM-CACGTAGGCGAACCC-MGBNFQ-3′) were used targeting IS*6110* (3), a 68-bp region of the transposon IS*6110*, specific for MTC pathogens. A QuantiFast Pathogen PCR + IC Kit (Qiagen, Hilden, Germany) was employed to evaluate each sample in duplicate in the MyiQ2 Two-Color real-time PCR Detection System (Bio-Rad, Hercules, CA, USA). A detection limit ranging from 10 to 100 genomic equivalents as well as a cut-off set at Cq <38 were previously described (27). Following the manufacturer's guidelines, an exogenous inhibition heterologous control (internal control assay) was included to evaluate the presence of certain inhibitors on the samples.

### Feces samples

For the aim of the present study, individual fresh feces samples from 31 cattle were collected at the slaughterhouse. All samples were collected directly from the distal portion of the rectum after the official routine veterinary examination and in accordance with national and European regulations.

Samples were directly collected into sterile containers and transported in refrigeration to the laboratory, where they were stored at −18°C until analysis. Sample processing before volatilome evaluation consisted of overnight thawing and weighting of 1.5 g of fresh feces in a precision balance on the day of analysis. This amount of sample was placed in a 20-mL glass headspace vial which was subsequently closed with a magnetic screw cap and a polytetrafluoroethylene (PTFE) silicone septum.

### Volatilome analysis and instrumentation

A FlavourSpec instrument (G.A.S. Gesellschaft für Analytics Sensorsysteme mbH, Dortmund, Germany) equipped with an autosampler (CTC Analytics AG, Zwingen, Switzerland) was employed for volatilome evaluation. Briefly, the samples inside glass headspace vials were incubated at 60°C for 15 min with agitation at 500 r.p.m. After

incubation, 100 µL of the headspace gas phase of the vial was automatically injected into the injector under splitless mode with the syringe of the autosampler, both at 80℃. Chromatographic separation was performed at 45℃ on a HP-5 capillary column 30 m × 0.32 mm ID × 0.5 µm film thickness (Agilent, Santa Clara, CA, USA). Nitrogen (99.999% purity) was used as the carrier gas under the following programmed flow: 1 mL/min for 13 min, 15 mL/min for 2 min, 20 mL/min for 13 min, after which the flow stopped. The analytes were ionized in the ion mobility spectrometry (IMS) ionization chamber, where a radiation energy of 6.5 KeV and 9.8 cm of drift tube were maintained at 75℃. The drift gas (nitrogen gas) was set at 150 mL/min. Every spectrum was recorded as the average scan of 12.

The integrated LAV (version 2.2.1) and VOCal (version 0.2.9) software were used for viewing and processing measurement data acquired by the instrument. The n-ketones C4–C9 (Sigma-Aldrich, Madrid, Spain) were employed as external references to calculate the retention index (RI) of VOCs under the same chromatographic conditions as the samples. Then, VOCs were identified by comparing the RI and drift time of standards in the GC-IMS Library (G.A.S. Gesellschaft für Analytics Sensorsysteme mbH).

## Chemometrics: data analysis

Considering the magnitude of GC-IMS data, the resulting information after feces analysis was treated by means of chemometrics. Specifically, a targeted approach was selected to process volatilome information. The preparation of data for its processing was as follows: the topographic maps generated by GC-IMS equipment were individually examined to select those available signals or markers. Afterwards, LAV software was employed to convert raw IMS data to .csv format, obtaining each individual signal (or feature) intensity. This step allowed data use and processing by MATLAB software (The Mathworks Inc., Natick, MA, USA, 2007) and its PLS Toolbox extension (Eigenvector Research, Inc., Manson, WA, USA) .

Subsequently, a principal component analysis (PCA) was initially performed in order to reduce data dimensionality and thus explore data, visualize potential trends and detect possible outliers. Data were preprocessed by autoscaling (mean centering and scaling to unit variance) previously to modeling. Classification models were based on the application of partial least squares-discriminant analysis (PLS-DA). For the calibration and cross-validation (CV; venetian blinds) of the models, the data set was split into a training set of 75% of the samples (calibration set) and a test set with the remaining 25% (validation set), which was used for the subsequent external validation (as blind samples). Considering the limited number of samples available for the present study, the sample set was randomly split three times, each with the validation and calibration percentages previously mentioned, to improve model robustness and accuracy. Moreover, the different splits between training and test sets were always done by keeping the ratio of samples as in the original set and balancing the representation of each group. The final number of latent variables selected in PLS-DA models was assessed by the minimum root mean square classification error in CV and prediction.

The variable importance for the projection (VIP) of every feature initially selected was also obtained to reduce data complexity and refine the final discriminant model, attempting to improve discrimination success by eliminating those features with a limited influence on model performance (VIP value <1) and keeping those considered significant (VIP value >1). VIP evaluation and the subsequent feature reduction were successively performed several times. Thus, sequential PLS-DA classification models were developed by selecting in each one the features with VIP value of ≥1, due to the VIPs being scaled in such a way that all compounds that have a VIP of ≥1 are considered relevant (29). Therefore, after the step of feature selection, PLS-DA models were recalculated, repeating the whole process (VIP evaluation, feature reduction, and model repetition) until the obtention of optimal results. The main goal was to achieve the greatest performance but without affecting the discriminant strength of models. This strategy was performed with the three sets of split samples.

The diagnostic performance of the present approach was evaluated by comparing with the microbiological culture. However, since the culture is considered an imperfect reference technique for MTC diagnosis (30), the statistical Epidat (version 3.1) software (Galician Health Service, Spain) was used to estimate the adjusted SE and SP.

Lastly, MetaboAnalyst (version 5.0) software was used to perform a volcano plot statistical analysis. This analysis combined the results from fold change analysis and $P$ value from $t$-test to visually identify those significant features based on their statistical significance as well as to select potential biomarkers of the two groups in comparison. This representation was performed with the autoscaled intensity from all the VOCs detected after GC-IMS analysis.

As mentioned above, VOC signals in the topographic plots were tentatively identified according to their retention and drift time using VOCal software and the available GC-IMS library (G.A.S. Gesellschaft für Analytics Sensorsysteme mbH).

## RESULTS

### Evaluation of topographic maps and feature selection

GC-IMS analysis of fecal samples resulted in two-dimensional-topographic plots where the different VOCs detected are characterized by the drift and retention times ($x$ and $y$ axes, respectively) and the intensity, attending to the signals' color and size. Fig. 1 shows a comparison between the topographic map of two fecal samples as examples: the first from an MTC-infected cow (Fig. 1A) and the second belonging to a non-infected individual (Fig. 1B). Some differences between the two samples can be visualized regarding the two topographic plots presented in the figure: the number, as well as the intensity of the signals, seems to be lower in the case of the infected animal (Fig. 1A), compared with the topographic plot of the negative animal (Fig. 1B). However, there were also some VOCs which showed a greater intensity in the first case (Fig. 1A).

After the visual examination of all the 31 topographic plots generated by GC-IMS analysis of feces, a total of 260 features were finally selected. Subsequently, the resulting data were evaluated through chemometrics to confirm the preliminary visual differences previously mentioned. The initial PCA, which was performed using the 31 samples analyzed, showed a slight trend of grouping according to infected and non-infected samples by means of PC1, although with some sample overlapping. It did not reveal any outliers within the entire set of samples (Fig. 2); therefore, volatilome information from all the 31 fecal samples evaluated by GC-IMS analysis was used to develop discriminant models.

### Discriminant models

The PLS-DA models developed in the present study with three calibration sets (a total of 72 samples) were used to classify samples of the corresponding three validation sets (a total of 21 samples) as blind samples (external validation). The confusion matrix as well as results related to the classification and diagnostic performance of the external validation and CV obtained in models are available in Table 1.

Concerning external validation results obtained with the initial PLS-DA models, only one sample from MTC-infected cattle (belonging to Set 2) was incorrectly classified as non-infected. Hence, an average percentage of classification error of 8.33% was obtained for this group. As with the infected group, only one sample from the group of negative animals (belonging to Set 3) was incorrectly classified inside the group of infected cattle, reaching an 11.11% mean classification error in this group. The correct category classification percentage of each set of samples is also collected in Table 1. Finally, the global percentage of correct classification of the model was calculated considering the above-mentioned mean correct classification percentage of each category, achieving 90.28%. Regarding the model diagnostic performance, SE and SP parameters are summarized in Table 1. Results of 91.66% and 99.99% were obtained for mean SE and SP, respectively. Individual SE and SP data of each split of samples are displayed in Table 1.

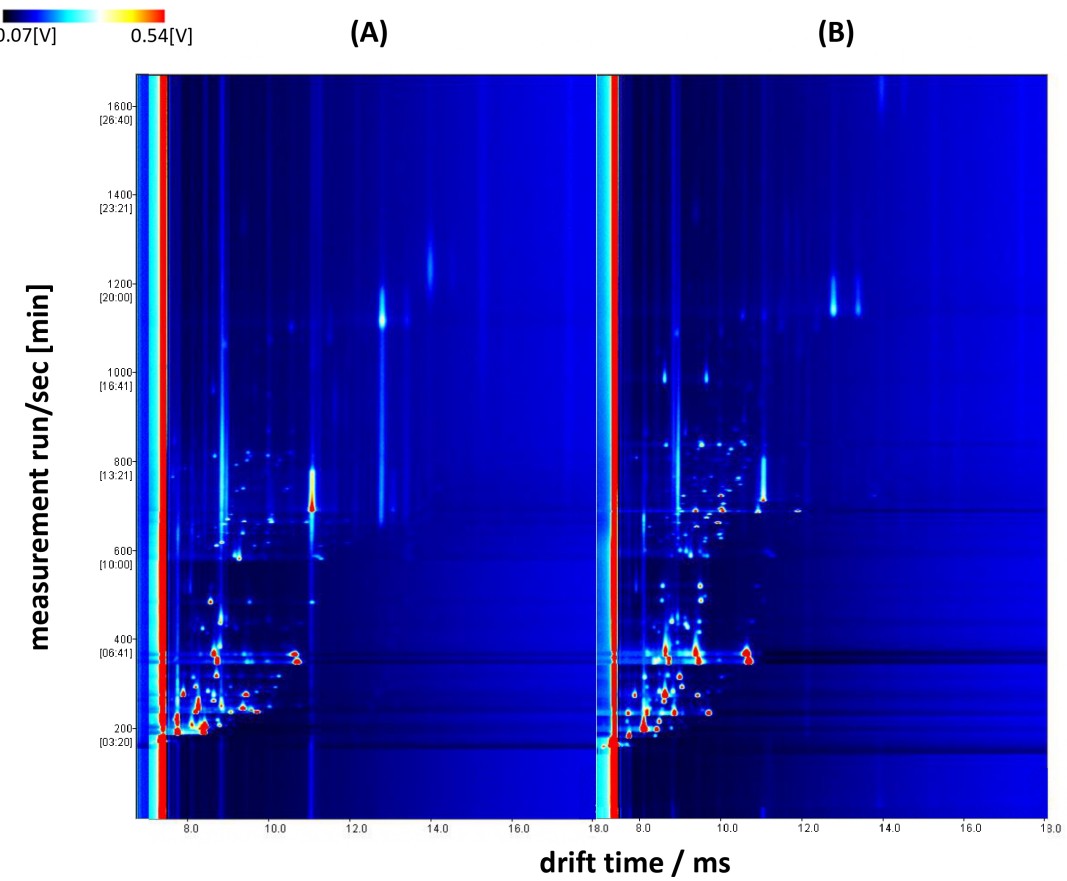

**FIG 1** Topographic plots obtained after feces analysis by GC-IMS for MTC-infected (A) and non-infected (B) cattle.

Despite the good classification and diagnostic performance obtained for external validation of discriminant models, the results of CV were quite different. In general, lower percentages of correct classification, SE and SP, were reached, ranging from 55% to 77%. Against this background, which reflects a low model robustness, VIP evaluation was subsequently performed in order to refine PLS-DA models and improve CV and external validation.

The score plots of the three PLS-DA models performed with Sets 1, 2, and 3 are available in Fig. 3. The resulting representation shows a clear separation between the groups of infected and non-infected animals. Nevertheless, a slight overlapping between groups is present in the score plot of Sets 2 and 3 (explaining the low classification rates obtained for CV; i.e., the low robustness of the models of these two sets): the spider representation design of score plots in Set 2 shows three infected samples close to the center of the non-infected group and one sample from the non-infected group close to the center of positive category. This is also in line with the classification results of external validation, since misclassified samples in confusion matrices belonged to these two sets (Sets 2 and 3).

## Model improvement by means of the evaluation of variable importance for the projection and selection of relevant compounds

VIP values of the 260 features employed for the PLS-DA model development were obtained for the three sets of samples. Subsequently, several reductions of features were applied considering the criteria explained in the methodology subsection (features whose VIP value was below 1 were discarded): for Set 1, four reductions of features were applied, obtaining seven features (VOCs numbered as 125, 140, 158, 175, 180, 191, and

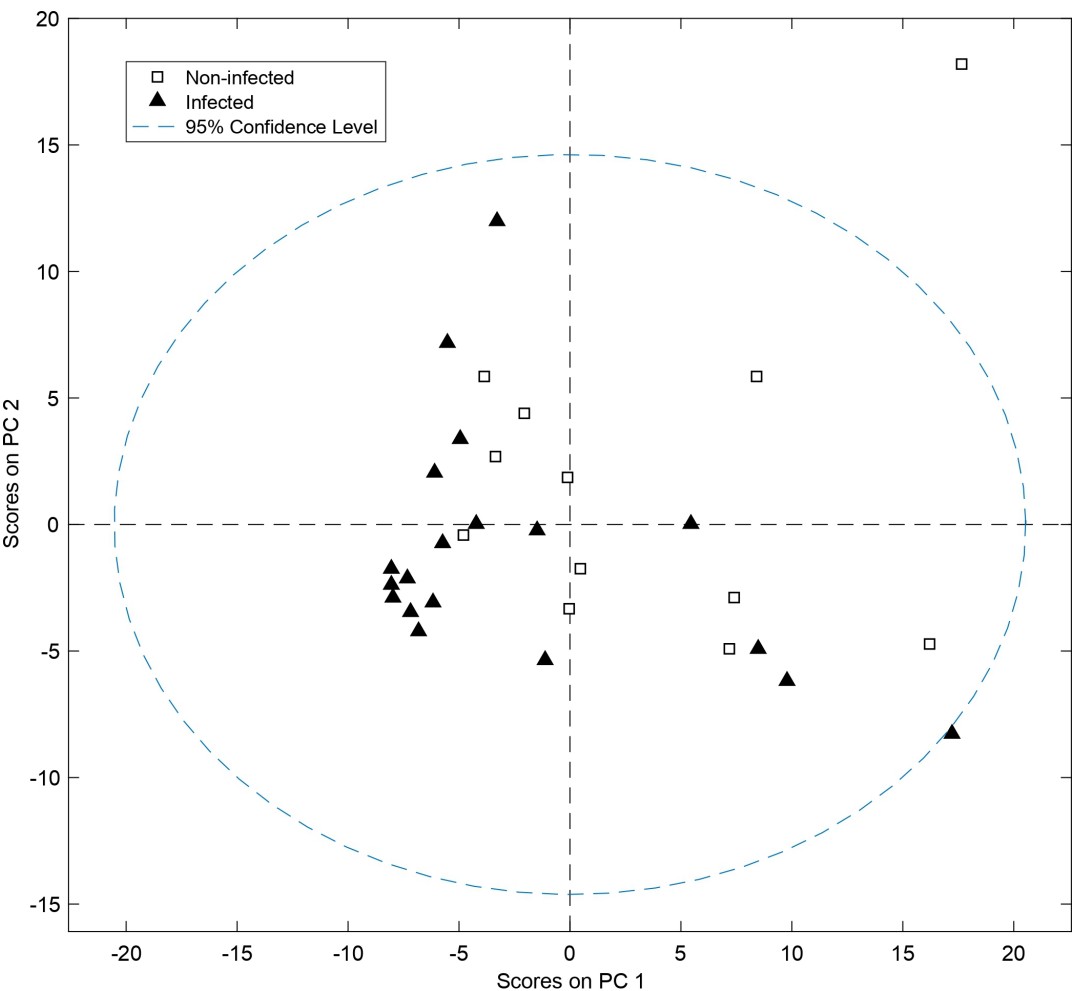

**FIG 2** PCA score plot obtained with volatilome information after GC-IMS analysis of feces.

**TABLE 1** Results of external and cross-validation (confusion matrices, classification performance, and diagnostic performance) obtained by the initial PLS-DA models developed with three different sets of samples (no VIP evaluation performed)

| | | | Cross-validation | | External validation | |
|---|---|---|---|---|---|---|
| | | | Real classification | | Real classification | |
| | | | Infected (Set 1/Set 2/Set 3) | Non-infected (Set 1/Set 2/Set 3) | Infected (Set 1/Set 2/Set 3) | Non-infected (Set 1/Set 2/Set 3) |
| Confusion matrices | Model prediction | Infected (Set 1/Set 2/Set 3) | 9/10/10 | 4/4/2 | 4/3/4 | 0/0/1 |
| | | Non-infected (Set 1/Set 2/Set 3) | 6/5/5 | 5/5/7 | 0/1/0 | 3/3/2 |
| | | Total number of samples (Set 1/Set 2/Set 3) | 15/15/15 | 9/9/9 | 4/4/4 | 3/3/3 |
| Classification performance | | Correct category classification percentage (Set 1/Set 2/Set 3) | 60.0/66.67/66.67 | 55.56/55.56/77.78 | 100.0/75.0/100.0 | 100.0/100.0/66.67 |
| | | Mean classification error (%) ± SD (%) | 35.55 ± 3.85 | 37.03 ± 12.83 | 8.33 ± 14.43 | 11.11 ± 19.24 |
| | | Total correct classification (%) ± SD[a] (%) | 63.71 ± 8.51 | | 90.28 ± 15.29 | |
| Diagnostic performance | | SE[a] (%) (Set 1/Set 2/Set 3) | 60.2/66.89/67.01 | | 99.99/75.0/99.99 | |
| | | SP[a] (%) (Set 1/Set 2/Set 3) | 69.35/75.26/99.1 | | 99.99/99.99/99.99 | |
| | | Mean SE[a] (%) ± SD[a] (%) | 64.70 ± 3.90 | | 91.66 ± 14.48 | |
| | | Mean SP[a] (%) ± SD[a] (%) | 81.24 ± 15.75 | | 99.99 ± 0.0 | |

[a]SD, standard deviation; SE, sensitivity; SP, specificity.

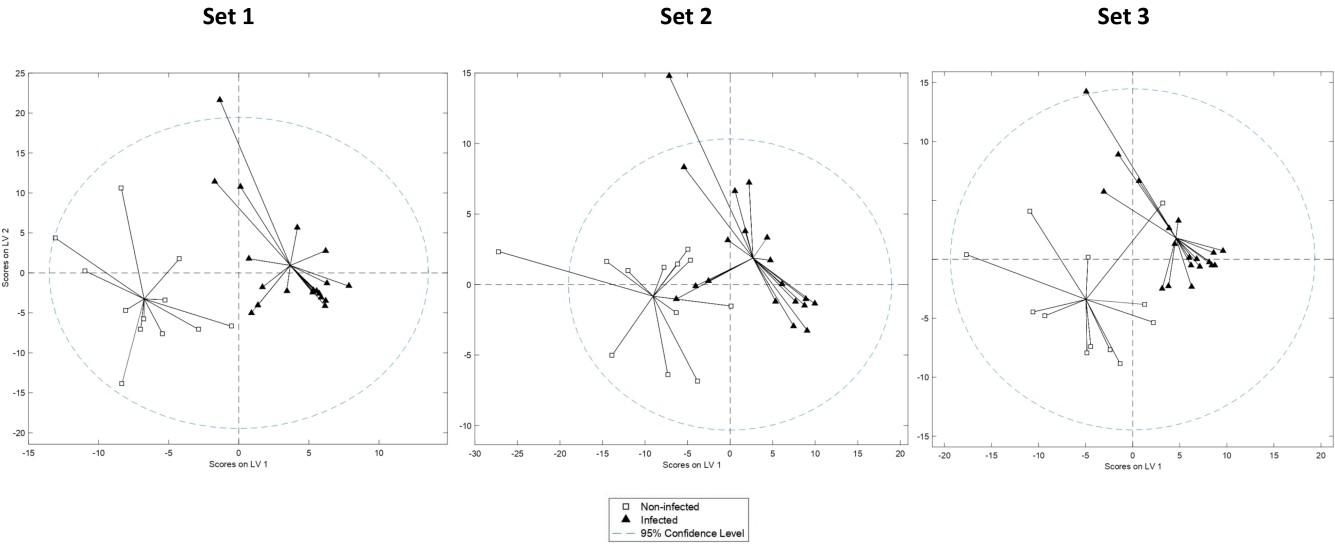

**FIG 3** Score plots of PLS-DA models performed using three different sets of samples.

220) as the definitive number for discriminant model development (the remaining 253 features were removed). In the case of Sets 2 and 3, this variable reduction process was repeated three times, deleting 243 and 250 features, respectively. The final numbers of features employed for PLS-DA model development in these sets were 17 for Set 2 (VOCs 16, 22, 29, 47, 56, 112, 125, 135, 152, 169, 180, 190, 191, 197, 198, 226, and 234) and 10 for Set 3 (VOCs 112, 119, 152, 158, 175, 179, 180, 191, 197, and 234). Figure 4 presents the whole process of VIP evaluation and feature reduction performed for Set 2 as an

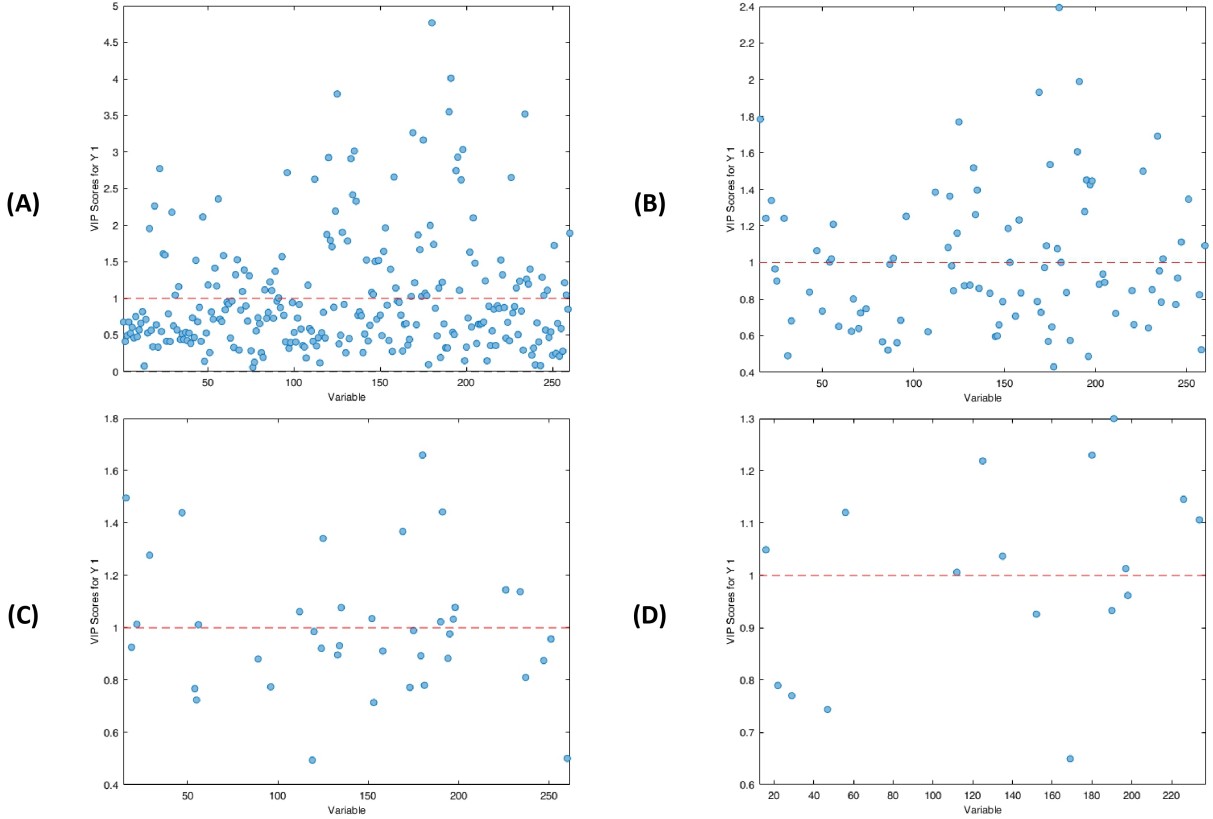

**FIG 4** Process of VIP evaluation and feature reduction performed for Set 2.

**TABLE 2** Results of external and cross-validation (confusion matrices, classification performance, and diagnostic performance) obtained by the final PLS-DA models developed with three different sets of samples (with VIP evaluation performed)

| | | | Cross-validation | | External validation | |
|---|---|---|---|---|---|---|
| | | | Real classification | | Real classification | |
| | | | Infected (Set 1/Set 2/Set 3) | Non-infected (Set 1/Set 2/Set 3) | Infected (Set 1/Set 2 /Set 3) | Non-infected (Set 1/Set 2/Set 3) |
| Confusion matrices | Model prediction | Infected (Set 1/Set 2/Set 3) | 14/15/14 | 1/2/1 | 3/4/4 | 0/0/0 |
| | | Non-infected (Set 1/Set 2/Set 3) | 1/0/1 | 8/7/8 | 1/0/0 | 3/3/3 |
| | | Total number of samples (Set 1/Set 2/Set 3) | 15/15/15 | 9/9/9 | 4/4/4 | 3/3/3 |
| Classification performance | | Correct category classification percentage (Set 1/Set 2/Set 3) | 93.33/100.0/93.33 | 88.89/77.78/88.89 | 75.0/100.0/100.0 | 100.0/100.0/100.0 |
| | | Mean classification error (%) ± SD[a] (%) | 4.45 ± 3.85 | 14.81 ± 6.41 | 8.33 ± 14.43 | 0.0 ± 0.0 |
| | | Total correct classification (%) ± SD[a] (%) | 90.37 ± 7.39 | | 95.83 ± 10.21 | |
| Diagnostic performance | | SE[a] (%) (Set 1/Set 2/Set 3) | 93.50/99.99/93.50 | | 75.0/99.99/99.99 | |
| | | SP[a] (%) (Set 1/Set 2/Set 3) | 99.99/99.99/99.99 | | 99.99/99.99/99.99 | |
| | | Mean SE[a] (%) ± SD[a] (%) | 95.66 ± 3.75 | | 91.66 ± 14.48 | |
| | | Mean SP[a] (%) ± SD[a] (%) | 99.99 ± 0.0 | | 99.99 ± 0.0 | |

[a]SD, standard deviation; SE, sensitivity; SP, specificity.

example. Figure 4A shows the initial 260 features, while Figure 4D presents the final 17 features employed for the definitive and refined model.

In general, the reduction of features after VIP evaluation resulted in a greater performance of classification models developed with Sets 1, 2, and 3 of samples. The results obtained for CV and external validation of the final PLS-DA models are summarized in Table 2. As can be seen, the general percentage of correct classification for CV and external validation was significantly higher (90.37% and 95.83%, respectively) than the data obtained for the initial models (63.71% and 90.28%, respectively) (Table 1). This was achieved since the classification error was significantly reduced. SE and SP parameters were also improved for CV results: 95.66% and 99.99%, respectively, which highlight a better robustness of models (Table 2). SE and SP achieved for external validation in model refinement was the same as those obtained before VIP evaluation: 91.66% and 99.99%, respectively, which reveal a good and constant performance of the final models after data reduction through VIP evaluation.

In general, the final number of VOCs selected after VIP evaluation for the three sets of samples was 23 (some of them were common to the three sets). Table 3 shows information about the intensity, standard deviation, *P* value and fold change result of this group of features. Furthermore, some of these above-mentioned 23 compounds were identified based on the searching results of the GC-IMS Library: dimer of acetic acid methyl ester (VOC 22), propanoic acid (VOC 47), 3-methylbutanal (VOC 119), 3-methyl-1-pentanol (VOC 158), pentenal (VOC 175), methyltrimethoxysilane (VOC 180), ethyl 2-methylpropanoate (VOC 190), thiophene (VOC 191), and hexanal (VOC 220). The information about this identification is also included in Table 3.

## Relevant fecal volatile organic compounds suggested by volcano plot in infected and non-infected cattle

In addition to the VIP evaluation, the importance of some of these VOCs was confirmed by another statistic test, the volcano plot, which highlighted the relevance of 26 VOCs detected after GC-IMS analysis of feces (Fig. 5). It was designed using results of fold change analysis and *t*-test of all the 260 VOCs initially obtained. This type of representation enabled discerning, after quick observation, those VOCs which obtained the greatest changes and were also significant. Three out of the 260 VOCs represented in the volcano plot (numbered as 220, 237, and 251) were significant for the group of

**TABLE 3** Information about the tentative identification, intensity, standard deviation, P value, and fold change result of significant VOCs pointed out after VIP evaluation and volcano plot representation

| VOC number | Identification | Infected Intensity[c] | Infected Standard deviation[c] | Non-infected Intensity[c] | Non-infected Standard deviation[c] | P | Fold change |
|---|---|---|---|---|---|---|---|
| 16[a] | – | 25.39 | 32.03 | 23.81 | 23.31 | 0.884 | 1.066 |
| 19[b] | Acetic acid methyl ester (monomer) | 24.68 | 19.56 | 51.08 | 32.12 | <0.01 | 0.48 |
| 22[a,b] | Acetic acid methyl ester (dimer) | 87.56 | 85.90 | 209.21 | 175.10 | 0.015 | 0.42 |
| 24[b] | – | 8.39 | 12.87 | 29.47 | 30.45 | 0.012 | 0.28 |
| 29[a] | – | 21.48 | 17.14 | 43.68 | 42.83 | 0.051 | 0.49 |
| 43[b] | – | 8.65 | 8.62 | 17.81 | 16.13 | 0.048 | 0.49 |
| 47[a,b] | Propanoic acid | 5.64 | 3.50 | 11.88 | 10.75 | 0.025 | 0.47 |
| 56[a,b] | – | 47.77 | 51.38 | 112.08 | 85.00 | 0.013 | 0.43 |
| 61[b] | – | 14.51 | 9.76 | 31.40 | 24.33 | 0.01 | 0.46 |
| 62[b] | 2-Hexanone | 30.97 | 26.11 | 78.79 | 74.70 | 0.015 | 0.39 |
| 64[b] | – | 39.23 | 33.94 | 85.47 | 59.17 | <0.01 | 0.46 |
| 67[b] | – | 13.75 | 6.04 | 29.83 | 23.30 | <0.01 | 0.46 |
| 73[b] | – | 8.96 | 4.62 | 37.23 | 56.41 | 0.036 | 0.24 |
| 83[b] | – | 6.83 | 3.85 | 14.28 | 14.00 | 0.035 | 0.48 |
| 112[a,b] | – | 22.68 | 15.47 | 71.91 | 59.21 | <0.01 | 0.32 |
| 119[a] | 3-Methylbutanal | 41.02 | 13.85 | 53.12 | 23.26 | 0.079 | 0.77 |
| 120[b] | Pentan-1-ol | 21.94 | 20.53 | 44.44 | 37.85 | 0.04 | 0.49 |
| 122[b] | Styrene | 9.57 | 5.34 | 19.49 | 13.91 | <0.01 | 0.49 |
| 125[a] | – | 7.89 | 5.23 | 57.73 | 60.74 | <0.01 | 0.14 |

| VOC number | Identification | Infected Intensity[c] | Infected Standard deviation[c] | Non-infected Intensity[c] | Non-infected Standard deviation[c] | P | Fold change |
|---|---|---|---|---|---|---|---|
| 135[a] | – | 7.31 | 5.66 | 10.13 | 6.13 | 0.202 | 0.72 |
| 140[a] | – | 10.90 | 17.68 | 12.00 | 8.35 | 0.842 | 0.91 |
| 145[b] | – | 6.33 | 4.48 | 20.95 | 22.98 | 0.01 | 0.30 |
| 152[a] | – | 13.08 | 4.43 | 20.71 | 8.78 | <0.01 | 0.63 |
| 156[b] | – | 5.37 | 1.54 | 11.08 | 10.57 | 0.027 | 0.48 |
| 158[a] | 3-Methyl-1-pentanol | 7.11 | 2.37 | 13.57 | 6.71 | <0.01 | 0.52 |
| 169[a] | – | 76.91 | 51.12 | 60.49 | 19.24 | 0.298 | 1.27 |
| 175[a,b] | Pentenal | 22.83 | 8.51 | 47.27 | 26.18 | <0.01 | 0.48 |
| 179[a] | – | 15.00 | 5.84 | 26.62 | 12.76 | <0.01 | 0.56 |
| 180[a,b] | Methyltrimethoxysilane | 13.22 | 8.66 | 33.79 | 23.17 | <0.01 | 0.39 |
| 190[a] | Ethyl 2-methylpropanoate | 95.57 | 109.93 | 41.16 | 41.59 | 0.113 | 2.32 |
| 191[a,b] | Thiophene | 19.84 | 6.60 | 51.76 | 27.13 | <0.01 | 0.38 |
| 197[a,b] | – | 5.32 | 2.02 | 13.22 | 11.50 | <0.01 | 0.40 |
| 198[a] | – | 6.95 | 5.65 | 8.48 | 3.94 | 0.42 | 0.82 |
| 220[a,b] | Hexanal | 26.32 | 15.90 | 12.77 | 4.19 | <0.01 | 2.06 |
| 226[a] | – | 45.32 | 52.26 | 17.85 | 4.78 | 0.081 | 2.54 |
| 234[a,b] | – | 17.08 | 12.64 | 81.28 | 75.40 | <0.01 | 0.21 |
| 237[b] | Propanal | 68.81 | 54.75 | 32.12 | 16.28 | 0.032 | 2.14 |
| 251[b] | – | 31.54 | 25.87 | 10.42 | 5.32 | <0.01 | 3.03 |

[a]Features pointed out as significant after VIP evaluation.
[b]Features pointed out as significant in volcano plot.
[c]Intensity and standard deviation data presented correspond to mean results of each group.

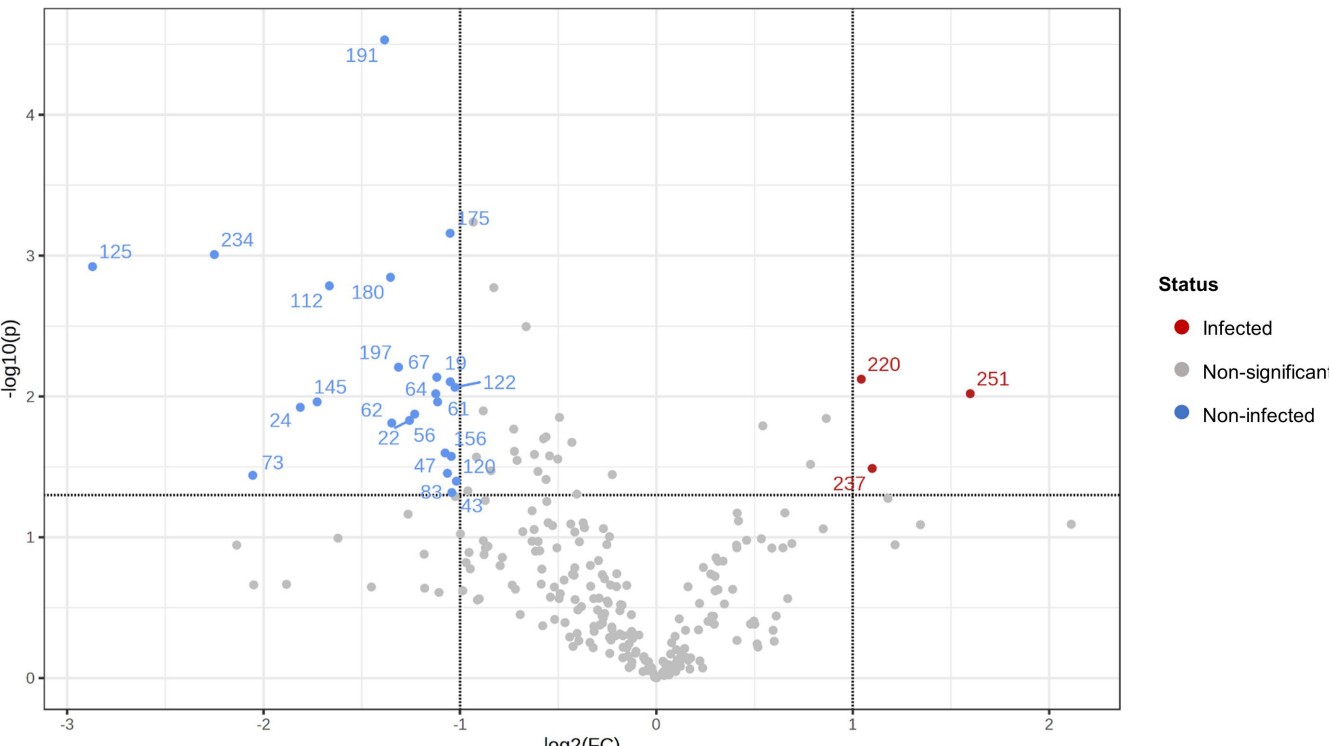

**FIG 5** Volcano plot designed with the intensity of the initial 260 VOCs detected in the GC-IMS analysis of feces. Colored dots correspond to those VOCs with a higher intensity in MTC-infected (red) and non-infected (blue) cattle.

MTC-infected animals, while 23 features (VOCs 19, 22, 24, 43, 47, 56, 61, 62, 64, 67, 73, 83, 112, 120, 122, 125, 145, 156, 175, 180, 191, 197, and 234) were associated with negative animals (Fig. 5). The VOCs numbered 220 and 251 were the most significant in MTC-positive animals, whereas the features 125, 175, 191, and 234 were the VOCs with the highest significance in the group of non-infected cattle.

The identification performed for the group of relevant compounds shown by volcano plot is also available in Table 3, together with the significant features obtained after VIP evaluation. On that list, 11 out of 26 VOCs were successfully identified: acetic acid methyl ester (monomer, VOC 19, and dimer, VOC 22), propanoic acid (VOC 47), 2-hexanone (VOC 62), pentan-1-ol (VOC 120), styrene (VOC 122), pentenal (VOC 175), methyltrimethoxysilane (VOC 180), and thiophene (VOC 191) in the group of significant compounds of non-infected cattle, and hexanal (VOC 220) and propanal (VOC 237) in the group of relevant features in MTC-infected animals. As can be seen, there was some consistency between the list of significant compounds highlighted by volcano plot and those selected after VIP evaluation. Furthermore, information about the intensity, standard deviation, P value, and results of fold change analysis from the 26 relevant VOCs above-mentioned is summarized in Table 3.

## DISCUSSION

VOCs can be directly analyzed from diverse biological matrices such as blood, serum, breath, feces, sweat, skin, and urine. However, cattle feces were chosen in the present study for several reasons. First, it is considered one of the most valuable sources of information about the physiology and metabolic state of an individual (20). Thus, alterations which affect any organic system may be reflected in fecal VOCs, a fact which is supported by different studies in both animal and human medicine. Moreover, feces also offer advantages related to the sampling process: immobilization or restraint of animals is not necessary, as samples can be directly collected after spontaneous defecation,

which is a process that eliminates the stress that this type of operation usually implies. Therefore, feces hold potential as a non-invasive diagnostic approach in livestock.

## Evaluation of topographic plots and feature selection

The visual differences highlighted in the present study between the two types of samples in the evaluation of the topographic plots were consistent with the subsequent results of discriminant models, which achieved high percentages of correct classification. The variations in terms of signal intensity as well as the absence or presence of several features would have enabled obtaining a successful differentiation of infected and non-infected animals. Information about 260 features was finally obtained due to the SE of GC-IMS, which has been underlined as an important advantage compared with other analytical techniques used for volatilome evaluation since low-concentration volatile compounds can be clearly detected (31, 32). The detection of such a high number of features in cattle feces is consistent with previous results using GC-IMS for volatilome analysis of animal feces: 200 VOCs in mice (33) and 409 features in pigs (15, 34), and 763 peaks in goats using differential IMS (35). This last study employed fecal VOC analysis to discriminate mycobacteria infection in goats. The above context gives coherence to the obtention of a great deal of interesting information (260 VOCs) from the analysis of fecal volatilome in cattle.

Although this methodology is not yet a widespread combination, it offers some advantages compared to other technologies: apart from the SE mentioned above, GC-IMS is operated under atmospheric pressure, making the adaptation of compact and portable options possible (36). This would be considered of paramount importance regarding the possible development of devices for pen-side use in the future. In addition, further research about volatilome persistence after spontaneous defecation in farms would be interesting to evaluate real-time monitoring methodologies.

## Discriminant models and variable importance for the projection

The evaluation of VIP performed in the present study for the three sets of samples enabled substantially reducing data complexity but also improving both the classification and diagnostic performance of discriminant models. These findings are in line with the extent of use given to VIP in the literature, as it has been shown to be useful for the selection of those features which are most relevant in terms of discrimination. The features or variables with a VIP value higher than 1 are considered to be significant for differentiation, while those whose VIP value is below 1 can be removed because of its scarce relevance (29). In this regard, more than 90% of features were excluded from discriminant models in the present study through VIP evaluation, later obtaining better classification results. This fact underlines the importance of variable selection for model performance; sometimes, the inclusion of a huge number of variables in discriminant models can prove ineffective and counterproductive, while the selection of a reduced number of variables but with significant relevance leads to a better performance. Therefore, the use of an appropriate chemometric data treatment, combined with an analytical methodology which provides substantial information, has been shown to be decisive in order to obtain satisfactory outcomes (34).

A scarce number of studies have been previously focused on the analysis of VOCs and its association with bTB infection in cows. Three studies have used exhaled breath as a source of volatilome information (37–39); two studies employed serum (14, 40); and only one study selected feces as a diagnostic matrix (41). However, the discrimination achieved in the above-mentioned studies was quite effective. In particular, the only work which employed fecal samples obtained SE and SP parameters that were very similar to our results: an SE ranging from 83% to 100% and an SP of 100% (41) vs 91.66% and 99.99% for SE and SP, respectively, in the present study. Hence, the results obtained here confirm the potential and the interest of volatilome analysis for the detection of natural or field infections by MTC in cattle.

## Relevant fecal volatile organic compounds in MTC-infected and non-infected cattle after VIP evaluation and volcano plot

Some differences in terms of number and intensity of VOCs between infected and non-infected samples were visually observed (Fig. 1), obtaining a higher intensity of features for the non-infected samples. This visual finding is consistent with the *P* values obtained in Table 3.

The VOCs listed in Table 3, which were obtained from two chemometric approaches (i.e., VIP evaluation and volcano plot), could be considered as potential markers of the presence or the absence of the disease. Moreover, a significant consistency was also obtained between the list of relevant VOCs selected in the present study and those compounds already pointed out in the literature as useful for mycobacterial infection detection in animals (24). Compounds such as the acetic acid methyl ester, 2-hexanone, styrene, and hexanal have been detected in previous studies which employed the analysis of VOCs from different biological matrices to discriminate different mycobacterial infections, including MTC, in goats, wild boars, or white-tailed deer (23, 25, 26).

In contrast, eight of the potential markers currently identified have not previously been associated with the discrimination of the infection by MTC in animals: propanoic acid, pentan-1-ol, pentenal, methyltrimethoxysilane, 3-methyl-1-pentanol, and 3-methylbutanal in the case of non-infected animals and propanal and ethyl 2-methylpropanoate as relevant for MTC-infected cows. This last group of two novel VOCs might be directly related to the infection by MTC as evaluated herein, but the first group is also considered important due to the possibility of VOC production, reduction, and inhibition associated with the infection. While the information already published by other authors comes mainly from studies with experimentally infected cattle, the present results were obtained after the evaluation of animals naturally infected with MTC. Therefore, the study of a natural infection may have revealed some differences and new VOCs associated with the process in comparison with assays conducted in controlled conditions. Our results underline the importance of validating data obtained from experimental infections in settings under field conditions.

The interest of these results is also connected with the context of volatilome analysis as an innovative approach for the detection of bTB infection, where the number of *in vivo* assays which attempt to optimize this methodology is considered very scarce (24). The only previous evaluation of fecal volatilome in cattle was carried out in controlled conditions with *M. bovis*-inoculated animals (41). Interestingly, both studies presented some coincidences: the classes of compounds known as alcohols and thiophenes were also detected in the present study after GC-IMS analysis of feces.

### General discussion

In addition to feces, other samples such as exhaled breath or serum have been previously employed for bTB infection detection through volatilome evaluation in cattle (24). However, some drawbacks, such as the stress induced, the complex equipment required for breath collection (modified masks, nostril samplers, or alveolar sampling devices), or the invasive character and the immobilization needed for serum obtention, have been highlighted. Therefore, this context emphasizes feces as a feasible sample for MTC infection detection in cattle through a volatilome analysis.

The present study is consistent with the methodology previously employed in investigations within the framework of volatilome analysis as a diagnostic methodology for mycobacteria infection. The number of animals included in this type of assay is usually small due to the complexity of the sampling methodology, the diagnostic reference necessarily demanded, or the need for specific facilities and equipment. In fact, the only study which has previously evaluated fecal volatilome from cattle to discriminate bTB infection included only 20 calves in the experimental design (41). Furthermore, the remaining studies performed with cattle in this research line but analyzing different biological samples are also characterized by a limited experimental set of animals: 21 and

27 cattle in the case of serum (14, 40), and 23 and 27 individuals when exhaled breath was used (37, 39).

## Conclusions

The present study has shown important differences in the VOCs produced in the headspace over fecal samples from MTC-infected and non-infected cattle. The combination between GC-IMS analysis and chemometric data treatment has been useful to obtain a high percentage of correct classification in discriminant models, optimal SE and SP parameters in the diagnostic performance, as well as a significative reduction in VOCs needed for the discrimination. Some of these compounds have also been pointed out as relevant after VIP evaluation and volcano plot for the discrimination between infected and non-infected animals. Our results also reveal some new VOCs which could be associated with the course of MTC infection. Against that background, future studies which increase the sample size would be of great benefit to confirm the potential of this approach as an *antemortem* diagnostic methodology for the detection of field infections by MTC, as well as to evaluate the repeatability of the group of new features highlighted. Furthermore, considering that the collection of feces consists of an easy and simple process which offers several advantages in comparison with other biological matrices, this study showed that it could be addressed as the sample of choice in future approximations of volatilome analysis, not only with cattle but also with other animal species.

## AUTHOR AFFILIATIONS

[1]Department of Animal Production, UIC Zoonosis y Enfermedades Emergentes ENZOEM, International Agrifood Campus of Excellence (ceiA3), University of Córdoba, Córdoba, Spain

[2]Department of Analytical Chemistry, International Agrifood Campus of Excellence (ceiA3), University of Córdoba, Córdoba, Spain

[3]Departamento de Nutrición y Bromatología, Área de Nutrición y Bromatología, Toxicología y Medicina Legal, Facultad de Farmacia, Universidad de Sevilla, Seville, Spain

[4]Department of Anatomy and Comparative Pathology and Toxicology, Pathology and Immunology Group (UCO-PIG), UIC Zoonosis y Enfermedades Emergentes ENZOEM, International Agrifood Campus of Excellence (ceiA3), University of Córdoba, Córdoba, Spain

[5]Department of Animal Health, UIC Zoonosis y Enfermedades Emergentes ENZOEM, International Agrifood Campus of Excellence (ceiA3), University of Córdoba, Córdoba, Spain

[6]Research and Agri-food Quality Centre (CICAP), Pozoblanco, Córdoba, Spain

## AUTHOR ORCIDs

Pablo Rodríguez-Hernández  http://orcid.org/0000-0001-6142-5975
María José Cardador  http://orcid.org/0000-0003-1859-9990
Rocío Ríos-Reina  http://orcid.org/0000-0002-9703-5853
José María Sánchez-Carvajal  http://orcid.org/0000-0003-0852-3765
Ángela Galán-Relaño  http://orcid.org/0000-0002-2665-1405
Francisco Jurado-Martos  http://orcid.org/0000-0002-1504-6293
Inmaculada Luque  http://orcid.org/0000-0003-1838-2636
Lourdes Arce  http://orcid.org/0000-0002-7130-8446
Jaime Gómez-Laguna  http://orcid.org/0000-0002-1787-5642
Vicente Rodríguez-Estévez  http://orcid.org/0000-0003-0148-2892

## FUNDING

| Funder | Grant(s) | Author(s) |
|---|---|---|
| Universidad de Córdoba (University of Cordoba) | Submodality 2.2 "Predoctoral research staff" | Pablo Rodríguez-Hernández |
| European Innovation Partnership For Agricultural Productivity and Sustainability | GOP2I-CO-16-0010 | Jaime Gómez-Laguna |

## AUTHOR CONTRIBUTIONS

Pablo Rodríguez-Hernández, Data curation, Formal analysis, Methodology, Validation, Writing – original draft | María José Cardador, Data curation, Methodology, Validation, Writing – review and editing | Rocío Ríos-Reina, Data curation, Validation, Writing – review and editing | José María Sánchez-Carvajal, Formal analysis, Investigation, Writing – review and editing | Ángela Galán-Relaño, Formal analysis, Investigation, Writing – review and editing | Francisco Jurado-Martos, Formal analysis, Investigation, Writing – review and editing | Inmaculada Luque, Investigation, Methodology, Supervision, Writing – review and editing | Lourdes Arce, Funding acquisition, Methodology, Supervision, Writing – review and editing | Jaime Gómez-Laguna, Conceptualization, Methodology, Project administration, Supervision, Writing – review and editing | Vicente Rodríguez-Estévez, Conceptualization, Funding acquisition, Methodology, Project administration, Writing – review and editing

## DATA AVAILABILITY

The raw data set was deposited at FigShare online open access repository with the doi number 10.6084/m9.figshare.23090465.

## ETHICS APPROVAL

No ethical or human consent was required as there was not any killing purpose of animals for the present study.

## ADDITIONAL FILES

The following material is available online.

Open Peer Review

**PEER REVIEW HISTORY (review-history.pdf).** An accounting of the reviewer comments and feedback.

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
