## [Reviewer comments · Microbiology Spectrum]

Microbiology Spectrum

Detection of *Mycobacterium tuberculosis* complex field infections in cattle using faecal volatile organic compound analysis through gas chromatography-ion mobility spectrometry combined with chemometrics

Pablo Rodríguez-Hernández, María José Cardador, Rocío Ríos-Reina, Jose Maria Sánchez-Carvajal, Ángela Galán-Relaño, Francisco Jurado-Martos, Inmaculada Luque Moreno, Lourdes Arce, Jaime Gómez-Laguna, and Vicente Rodríguez-Estévez

Corresponding Author(s): Jaime Gómez-Laguna, Universidad de Cordoba

Review Timeline:

Submission Date:	May 2, 2023
Editorial Decision:	May 17, 2023
Revision Received:	June 14, 2023
Accepted:	June 30, 2023

Editor: Florence Doucet-Populaire

Reviewer(s): Disclosure of reviewer identity is with reference to reviewer comments included in decision letter(s). The following individuals involved in review of your submission have agreed to reveal their identity: Christian Gortázar (Reviewer #1); La'Tonzia L Adams (Reviewer #3)

Transaction Report:

DOI: <https://doi.org/10.1128/spectrum.01743-23>

May 17, 2023

Dr. Jaime Gómez-Laguna
Universidad de Cordoba
Anatomy and Comparative Pathology
Campus de Rabanales. Edificio Sanidad Animal
Ctra Madrid-Cádiz, km 396 s/n
Córdoba, Córdoba 14014
Spain

Re: Spectrum01743-23 (Detection of *Mycobacterium tuberculosis* complex field infections in cattle using faecal volatile organic compound analysis through gas chromatography-ion mobility spectrometry combined with chemometrics)

Dear Dr. Jaime Gómez-Laguna:

Link Not Available

Sincerely,

Florence Doucet-Populaire

Journals Department
Reviewer comments:

Reviewer #1 (Public repository details (Required)):

Spectroscopy outputs should be deposited

Reviewer #1 (Comments for the Author):

Main comments

- 1.- The introduction lacks a statement of the study goals at the end. You make the case that this technology has potential to contribute to improved TB diagnosis, but please clearly state the hypothesis and specific goals of this experiment.
- 2.- Methods: please specify from how many farms came the sampled animals and if the negative controls came from farms which also provided positive animals. This will not change the results and their interpretation, but would eventually imply that a better sampling (taking negative controls from confirmed-negative farms) could have improved diagnostic accuracy.
- 3.- In the discussion on future pen-side uses, a comment of volatilome persistence after feces are deposited would be opportune. Ideally, this could be an additional experiment to perform under controlled conditions both in infected and uninfected cattle.

Minor comments and suggested edits

Line 60 M bovis infection

L71 30% sounds very high, please confirm.

L87 skin testing does not follow this... it doesn't require special facilities or skills - please edit

L101 a reference or two would be needed at the end of this statement

L109 identified instead of addressed?

L270-271 please state the actual numbers and proportions rather than using vague expressions such as "all except one" or similar. This should be corrected throughout the results section.

Reviewer #3 (Comments for the Author):

I think this is quite a novel method of assessing for MTC infections. However, increasing the sample size of the study would be of great benefit.

Staff Comments:

Preparing Revision Guidelines

Please return the manuscript within 60 days; if you cannot complete the modification within this time period, please contact me. If you do not wish to modify the manuscript and prefer to submit it to another journal, please notify me of your decision immediately so that the manuscript may be formally withdrawn from consideration by Microbiology Spectrum.

Paper #Spectrum01743-23. Comments to the reviewer

Firstly, we would like to thank the suggestions and comments from reviewers. We fully agree and think that these will help to improve the quality of the manuscript. The changes made in the manuscript are marked with red (and strike out) and green colors.

Also, minor changes in Tables 1 and 2, and a modification due to a mistake (Line 318: change “251” by “250”) have been introduced.

Reviewer 1

1. Public repository details

1.1 “Spectroscopy outputs should be deposited”.

- The raw data from the study has been deposited in the online open access repository and a new sentence has been included at this regard as follows (Line 248-250): “2.5. Data availability. The raw data set was deposited at FigShare online open access repository with the doi number 10.6084/m9.figshare.23090465.”

2. Comments for the Author

2.1 “The introduction lacks a statement of the study goals at the end. You make the case that this technology has potential to contribute to improved TB diagnosis, but please clearly state the hypothesis and specific goals of this experiment.”.

The end of the introduction has been modified to clearly present the specific goal of the study.

Changes:

- Lines 113-116: change “Thus, considering the benefits of this approach as well as the advantages that it would provide to the current diagnostic panorama of mycobacteria infection in livestock, the present study seeks to improve the state of knowledge and the number of *in vivo* assays of this VOC analysis application.” by “Thus, considering the benefits of this approach as well as the advantages that it would provide to the current diagnostic panorama of mycobacteria infection in livestock, the improvement of the state of knowledge and the number of *in vivo* assays of this VOC analysis application is considered important.”
- Lines 119-122: change “For this purpose, the GC-IMS technique together with chemometrics have been employed to discriminate naturally infected cattle by MTC from non-infected animals.” by “In this regard, the goal of the present study was to evaluate and validate the combination of the GC-IMS technique and chemometrics to discriminate naturally infected cattle by MTC from non-infected animals.”

- Line 122: adding “For this purpose, [...]” at the beginning of the sentence.

2.2 “Methods: please specify from how many farms came the sampled animals and if the negative controls came from farms which also provided positive animals. This will not change the results and their interpretation, but would eventually imply that a better sampling (taking negative controls from confirmed-negative farms) could have improved diagnostic accuracy.”

Methods section has been revised to include the information suggested by reviewer 1. Although the number of farms was already presents in the manuscript, additional information about the origin of negative animals has been included.

Changes:

- Lines 135-136: adding “Although the majority of non-infected animals were reared in herds with an officially tuberculosis-free status, some of them came from positive farms.”

2.3 In the discussion on future pen-side uses, a comment of volatilome persistence after feces are deposited would be opportune. Ideally, this could be an additional experiment to perform under controlled conditions both in infected and uninfected cattle.

We agree with the idea shared by reviewer 1. Therefore, a new sentence has been included to present that information.

Changes:

- Lines 405-406: adding “In addition, further research about volatilome persistence after spontaneous defaecation in farms would be interesting to evaluate real-time monitoring methodologies.”

3. Minor comments and suggested edits

3.1 Line 60 M bovis infection: We appreciate the suggestion, but no more keywords are allowed by the journal.

3.2 Line 71: 30% sounds very high, please confirm.: Information and data have been revised as proposed by reviewer 1. Although it sounds a high percentage, 30% is correct.

3.3 Line 87: skin testing does not follow this... it doesn't require special facilities or skills - please edit: That part of the manuscript is referred to the current bTB diagnosis, in general (as mentioned at the beginning of the sentence). We do not refer exclusively to skin test; we also refer to bacteriological culture, molecular techniques such as PCR,

indirect approaches such as ELISA... etc. Therefore, we consider that the drawbacks highlighted in that sentence should be included.

3.4 Line 101: a reference or two would be needed at the end of this statement: The suggestion of reviewer 1 has been considered and two references have included at the end of that sentence. Changes:

- Line 100: adding “(Nol et al., 2020; Spooner et al., 2009)”.

3.5 Line 109: identified instead of addressed?: The change suggested by reviewer 1 has been introduced. Change:

- Line 109: change “addressed” by “identified”.

3.6 Lines 270-271, please state the actual numbers and proportions rather than using vague expressions such as "all except one" or similar. This should be corrected throughout the results section. We agree with the recommendation of reviewer 1 and results section has been revised to avoid vague expressions. Changes:

- Lines 278-280: change “Concerning external validation results obtained with the initial PLS-DA models, all the samples from MTC-infected cattle were correctly classified or predicted, except for one sample from Set 2, which was incorrectly classified as non-infected.” by “Concerning external validation results obtained with the initial PLS-DA models, only one sample from MTC-infected cattle (belonging to Set 2) was incorrectly classified as non-infected.”
- Line 263: adding “31”.
- Line 266: delete “all” and adding “31”.
- Line 310: change “all” by “the 260”.
- Lines 353-354: “healthy” has been deleted.

Reviewer 3

1. Comments for the Author.

1.1 I think this is quite a novel method of assessing for MTC infections. However, increasing the sample size of the study would be of great benefit.

We appreciate the comment from reviewer 2 and although we agree, the samples used for the present study are part of a finished project about bTB surveillance and monitoring in the framework of the Spanish national eradication programme. Therefore, to include a higher number of samples is not feasible for this study although we will consider it for future studies.

However, that recommendation has been introduced in the conclusions:

- Lines 504-508: change “Against that background, future studies should consider this group of features to evaluate the repeatability and therefore, its potential

use as an *antemortem* diagnostic methodology.” by “Against that background, future studies which increase the sample size would be of great benefit to confirm the potential of this approach as an *antemortem* diagnostic methodology for the detection of field infections by MTC, as well as to evaluate the repeatability of the group of new features highlighted”.

June 30, 2023

Dr. Jaime Gómez-Laguna
Universidad de Cordoba
Anatomy and Comparative Pathology
Campus de Rabanales. Edificio Sanidad Animal
Ctra Madrid-Cádiz, km 396 s/n
Córdoba, Córdoba 14014
Spain

Re: Spectrum01743-23R1 (Detection of *Mycobacterium tuberculosis* complex field infections in cattle using faecal volatile organic compound analysis through gas chromatography-ion mobility spectrometry combined with chemometrics)

Dear Dr. Jaime Gómez-Laguna:

Your manuscript has been accepted, and I am forwarding it to the ASM Journals Department for publication. You will be notified when your proofs are ready to be viewed.

Sincerely,

Florence Doucet-Populaire
Editor, Microbiology Spectrum
